# A Comprehensive Analysis of Auxin Response Factor Gene Family in *Melastoma dodecandrum* Genome

**DOI:** 10.3390/ijms25020806

**Published:** 2024-01-09

**Authors:** Yukun Peng, Kai Zhao, Ruiyue Zheng, Jiemin Chen, Xuanyi Zhu, Kai Xie, Ruiliu Huang, Suying Zhan, Qiuli Su, Mingli Shen, Muqi Niu, Xiuming Chen, Donghui Peng, Sagheer Ahmad, Zhong-Jian Liu, Yuzhen Zhou

**Affiliations:** 1Ornamental Plant Germplasm Resources Innovation & Engineering Application Research Center, Key Laboratory of National Forestry and Grassland Administration for Orchid Conservation and Utilization, College of Landscape Architecture and Art, Fujian Agriculture and Forestry University, Fuzhou 350002, China; pengyukun@fafu.edu.cn (Y.P.); ruiyuezheng@fafu.edu.cn (R.Z.); jieminchen@fafu.edu.cn (J.C.); xuanyizhu@fafu.edu.cn (X.Z.); kai0526@fafu.edu.cn (K.X.); ruiliuhuang@fafu.edu.cn (R.H.); suyingzhan@fafu.edu.cn (S.Z.); qiulisu@fafu.edu.cn (Q.S.); muqiniu@fafu.edu.cn (M.N.); chenxiuming@fafu.edu.cn (X.C.); fjpdh@fafu.edu.cn (D.P.); sagheerhortii@gmail.com (S.A.); 2College of Life Sciences, Fujian Normal University, Fuzhou 350117, China; zhaokai@fjnu.edu.cn (K.Z.); qsx20221250@student.fjnu.edu.cn (M.S.)

**Keywords:** Myrtales, protein structure, miRNA regulation, gene expression, IAA response and inhibition

## Abstract

Auxin Response Factors (ARFs) mediate auxin signaling and govern diverse biological processes. However, a comprehensive analysis of the ARF gene family and identification of their key regulatory functions have not been conducted in *Melastoma dodecandrum*, leading to a weak understanding of further use and development for this functional shrub. In this study, we successfully identified a total of 27 members of the ARF gene family in *M. dodecandrum* and classified them into Class I–III. Class II–III showed more significant gene duplication than Class I, especially for *MedARF16s.* According to the prediction of cis-regulatory elements, the AP2/ERF, BHLH, and bZIP transcription factor families may serve as regulatory factors controlling the transcriptional pre-initiation expression of *MedARF*. Analysis of miRNA editing sites reveals that miR160 may play a regulatory role in the post-transcriptional expression of *MeARF*. Expression profiles revealed that more than half of the *MedARFs* exhibited high expression levels in the stem compared to other organs. While there are some specific genes expressed only in flowers, it is noteworthy that *MedARF16s*, *MedARF7A*, and *MedARF9B*, which are highly expressed in stems, also demonstrate high expressions in other organs of *M. dodecandrum*. Further hormone treatment experiments revealed that these *MedARFs* were sensitive to auxin changes, with *MedARF6C* and *MedARF7A* showing significant and rapid changes in expression upon increasing exogenous auxin. In brief, our findings suggest a crucial role in regulating plant growth and development in *M. dodecandrum* by responding to changes in auxin. These results can provide a theoretical basis for future molecular breeding in Myrtaceae.

## 1. Introduction

Auxin, being the predominant hormone in plants, assumes a pivotal role in governing various aspects of plant growth and development by promoting or inhibiting the synthesis of target gene products [1]. The Auxin Response Factor Gene Family was first identified in the model plant *Arabidopsis thaliana* [2]. Recent studies have provided compelling evidence highlighting the critical involvement of *ARFs* in signal transduction and growth development across diverse plant species [3]. They exhibit a remarkable ability to specifically bind to auxin response elements, thereby exerting regulatory control over auxin-responsive genes in plants [4].

Typical members of the ARF gene family are characterized by the presence of three conserved domains. First, the B3 DNA-binding domain is located at the N-terminus. Second, the middle region contains the Aux/IAA domain, which determines the transcriptional activation or inhibition activity of ARFs. Additionally, some ARF genes possess a C-terminal PBI domain [5]. The B3 DNA-binding domains (DBDs) are plant-specific B3-type domains composed of the B3 domain, dimerization domain, flanking domain, and Tudor-like auxiliary domain within the C-terminal region [6]. The B3 domain possesses DNA-specific binding activity and regulates gene expression by directly binding to the TGTCTC motif on the downstream auxin response element (AuxRE) in the promoter region [7]. The PBI domain, also known as the Auxin resp III or Auxin resp IV domain [8], displays a high degree of homology with domains III and IV of the Aux/IAA protein [9]. This specific region of the ARF protein facilitates interactions with Aux/IAA proteins and enhances DNA binding by promoting ARF dimerization [10].

The functions of ARFs have been extensively studied in the model plant *A. thaliana*. ARF genes exhibit distinct expression patterns across various tissues, organs, and developmental stages in different plant species, indicating differential regulation for each member [11]. For example, in *A. thaliana*, *AtARF1* and *AtARF2* are expressed in seedlings, roots, anthers, filaments, siliques, and senescent leaves [12]. *AtARF3* and *AtARF4* are expressed in reproductive and vegetative organs [13]. *AtARF5*, *6*, *7*, and *19* are expressed in developing flowers, young seedling roots, and developing embryos, where they regulate embryonic pattern formation [14], vascular tissue formation, flower maturation, and hypocotyl formation. In addition to *A. thaliana*, the expression of ARF genes has also been studied in other plants. For example, *OsARFs* in rice show varying degrees of expression in roots, stems, leaves, young panicles, and leaf sheaths [15]. *LcARFs* in *Litchi chinensis* are expressed in various developmental processes, particularly during flower formation and fruit abscission [16]. Furthermore, the ARF gene family has been studied at the whole-genome level in many other plant species [7], and a large number of results have demonstrated the high diversity and abundance of ARF genes in different plant species [17,18,19,20,21]. Therefore, in-depth studies on plant ARF transcription factors contribute to a more comprehensive understanding of the growth and developmental mechanisms in plants mediated by auxin [22].

*Melastoma dodecandrum*, belonging to the family Melastomataceae and the genus *Melastoma*, is a plant known for its vibrant flower color, nutritious fruit, strong stress tolerance, and dense growth. It can survive under relatively low water and nutrient conditions, making it an ideal ground cover plant on poor lands and forest slopes [23,24]. However, there has been limited research on the molecular mechanisms underlying the regulation of its growth and development. In recent years, the Ornamental Plant Germplasm Resources Innovation & Engineering Application Research Center at Fujian Agriculture and Forestry University has made significant progress in the assembly and annotation of *M. dodecandrum*’s nuclear genome [25], mitochondrial genome [26], and chloroplast genome [27]. This progress has allowed us to gain valuable insights into the genetic makeup and organization of *M. dodecandrum*. Additionally, our research has led to the identification and analysis of several key gene functions [28,29], shedding light on the molecular mechanisms underlying important traits in *M. dodecandrum*. Building on our previous findings, the present study utilizes genomic data of *M. dodecandrum* to identify and analyze the structural characteristics and expression patterns of ARF genes. This research aims to further explore the molecular functions of these special transcription factors during the growth and development of *M. dodecandrum*.

## 2. Results

### 2.1. Genome-Wide Identification of MedARFs

Using the genomic data of *M. dodecandrum*, a blast search was conducted in the *M. dodecandrum* database, using AtARF proteins as a reference, to identify the *MedARF* genes. The *M. dodecandrum* genome was also screened using the HMM to identify the ARF transcription factors. Finally, the NCBI CD-Search Tool was applied to filter the identification results, eliminating protein sequences with incomplete or redundant domains. As a result, we obtained a total of 27 members belonging to the ARF gene family, all of which possessed both the B3 and ARF domains. Based on the clustering of selected MedARF proteins and AtARF proteins in the evolutionary tree (Figure 1A), they were named MedARF2 to MedARF19, and we analyzed their physicochemical properties (Appendix A). These characteristics included the gene name, protein length, predicted isoelectric point (pI), molecular weight (MW), instability index (II), aliphatic index (AI), and grand average of hydropathicity (GRAVY). The MedARF proteins exhibited amino acid ranges spanning from 319 to 1148. The molecular weights of the corresponding proteins varied from 35.89 to 127.78 kDa, reflecting the proportional relationship with the number of amino acids. Notably, MedARF7B displayed the highest molecular weight, while MedARF5A exhibited the lowest. The theoretical isoelectric points of MedARF proteins ranged from 5.75 to 9.55. It is worth mentioning that MedARF5A, MedARF10A, MedARF10B, MedARF16A, MedARF16B, and MedARF16C displayed isoelectric points greater than 7.5, indicating an alkaline tendency. The aliphatic index ranged from 66.89 to 80.39, and the hydrophobicity index ranged from −0.567 to −0.286. Based on subcellular localization predictions, the majority of MedARF proteins were primarily localized in the cell nucleus. However, MedARF7B was located in the plasma membrane, and the MedARF2, MedARF17A, and MedARF17B were located in the chloroplast. Additionally, this study identified additional MedARF transcription factors that can simultaneously localize to the cell nucleus and cytoplasm. These included MedARF5B, MedARF6A, MedARF9A, MedARF16A, MedARF16B, and MedARF16C.

### 2.2. Phylogenetic and Tertiary Structure Analysis of MedARFs

*MedARFs*, with their orthologs from *A. thaliana*, *L. chinensis*, and *O. sativa*, were utilized to construct a phylogenetic tree for the classification of MedARF proteins. The phylogenetic analysis divided the 27 MedARF proteins into three distinct subfamilies: class I, II, and III, which contained 6, 9, and 12 members, respectively (Figure 1A). Based on the clustering results, MedARF3, MedARF4, and MedARF9 clustered together in Class I, while MedARF5s, MedARF6S, MedARF7s, and MedARF19 clustered together in Class II. The remaining genes clustered in the third branch. The consistency between the evolutionary tree and previous studies’ clustering results confirms the accuracy and reliability of the phylogenetic tree [16]. Subsequently, a detailed prediction and analysis of the tertiary structure of each branch of MedARF proteins in *M. dodecandrum* are depicted in (Figure 1B). The proportions of alpha helix, extended strand, beta turn, and random coil varied greatly among proteins of different branches. But the clustered proteins in the phylogenetic tree had similar tertiary structures. Interestingly, the tertiary structure of MedARF5A differed significantly from that of the other proteins in the first branch. Based on this, we compiled and analyzed the spatial structure occupancy ratios of all the MedARF proteins. The results can be found in (Appendix A). The results show that differences in length between MedARF5A and other Class I proteins may contribute to the observed differences in their tertiary structures. In addition, the B3 and Auxin resp domains of MedARF7A, MedARF4, and MedARF17A at the base of the three branches were extracted, respectively. The tertiary structure of MedARF proteins was predicted as a reference for the tertiary structure of each domain. The results (Appendix A) show that each ARF domain has one alpha spiral and four beta spirals, and each Auxin resp domain has two alpha spirals and seven beta spirals. It can be seen that the two domains have similar structures even in different branches, indicating that the ARF domain and Auxin resp domain in *MedARF* are relatively conservative in structure.

### 2.3. Analysis of Gene Structure and Conserved Motifs

According to the phylogenetic tree and comparing protein motifs and domains, 27 *MedARFs* were classified into three groups, consisting of nine, six, and twelve members, respectively (Figure 2A). Significant differences were observed in the gene structures of *MedARFs*, with 27 family members having multiple CDs, while the remaining members had only 1 to 4 CDs. In the MedARF protein sequences, 15 conserved motifs were identified (Figure 2E), and the codon bias of each motif is provided in the Appendix A. In addition, this study also selected MedARF7A at the bottom of the phylogenetic tree and extracted its ARF domain and Auxin resp domain separately. The tertiary structure of B3 and ARF domains of the MedARF7A protein was predicted as a reference. The results are labelled in Figure 2B,C. Motifs 1, 2, 3, 4, 5, 6, 7, and 8 were composed of B3 and ARF domains. Within the B3 domain, motif 1 contained β3, β4, and β5; motif 3 contained β1 and β2; motif 5 contained β6 and α2-helix; and motif 13 contained β7 and the C-terminus of the B3 domain (Figure 2C). In the ARF domain, motif 4 represented the C-terminus of the ARF domain; motif 7 contained β3 and β4; and motif 11 contained β1, β2, and α1-helix (Figure 2B). Certain motifs were specific to particular groups of the MedARF proteins. For instance, motifs 14 and 15 only appeared in Class III, while seven family members identified in the Class I and II groups had the PB1 motif (Figure 2E). The multiple sequence alignment analysis of the B3 and ARF domains showed significant similarity in the amino acid sequences of the *MedARFs* for improved clarity. The B3 domain contained 19 completely conserved amino acids, including 3-K, 5-L, 8-S, 9-D, 16-F, 17-S, 23-A, 28-P, 30-L, 37-P, 45-D, 51-W, 53-F, 55-H, 58-R, 70-W, 73-F, 79-L, and 98-G. Similarly, the ARF domain had four complete conserved amino acids, including 4-F, 10-P, 54-G, and 67-W.

### 2.4. Chromosomal Locations and Duplication Events

In order to analyze the occurrence of gene duplications among *MedARF* genes, we first performed a chromosomal localization of the identified 27 *MedARFs* in *M. dodecandrum*. There was a significant variation in the number of genes among different chromosomes. The *MedARF* genes were mainly located on chromosomes 1, 2, and 3, containing 5, 7, and 5 genes, respectively. No *MedARFs* were found on chromosomes 4, 5, 10, and 11. It is worth noting that one of the *MedARFs*, *MedARF9A*, was located on Contig13. Furthermore, we discovered 13 *MedARF* instances of segmental duplications among the twelve chromosomes, along with 2 instances of tandem duplications occurring within the same chromosome (Figure 3). Segmental duplications, which involve the duplication of large genomic segments, can lead to the expansion of gene families and the generation of new functional genes. Tandem duplications refer to the presence of multiple copies of a gene in close proximity to each other on the same chromosome. These duplications involve all genes except for *MedARF2*, *MedARF3*, *MedARF4*, *MedARF6A*, and *MedARF19*. Notably, both instances of tandem gene duplications involving *MedARF7A*, *MedARF7B*, *MedARF16C*, and *MedARF16* occurred on the second chromosome of *M. dodecandrum*, suggesting a common ancestor for the duplicated genes with similar functions and structures.

### 2.5. The cis-Regulatory Elements and Targets of Specific miRNAs for MedARF Genes

In this study, we used PlantPAN to predict cis-acting elements in the upstream 2000 bp region of the identified ARF genes in *M. dodecandrum*. The results are shown in Appendix A. A total of 11,806 unique transcription factor binding sites belonging to 37 different transcription factor families were identified within the upstream 2000 bp region of the *MedARF* gene promoters, exhibiting an uneven distribution among the 27 ARF genes, with varying numbers of transcription factor binding sites identified for each gene. The upstream regions of the *MedARF* gene family were enriched with cis-regulatory elements associated with the AP2/ERF, BHLH, bZIP, GATA, and WRKY transcription factor families (Figure 4A). Multiple ARF genes showed an abundance of cis-regulatory elements associated with the AP2/ERF, BHLH, bZIP, GATA, and WRKY transcription factor families within the first 35 bp, consistent with the predictions in the upstream 2000 bp region.

In addition to the cis-regulatory analysis, we also predicted miRNAs targeting the *MedARF* genes. MiRNA splicing is an important gene regulatory mechanism that regulates gene expression by binding to the mRNA of target genes and inducing their degradation or inhibiting their translation. We selected 27 members of the *MedARFs* as candidate target genes to predict the miRNA targeting these genes. Except for *MedARF2*, *MedARF3*, *MedARF5A*, *MedARF9A*, *MedARF9B*, *MedARF9C*, and *MedARF19*, the remaining members were predicted to have at least one miRNA binding to their transcripts (Appendix A). Interestingly, 38 miR160 editing sites were predicted in *MedARFs*, and miR160b binding sites with binding abundance were predicted in the *MedARF16C*, *MedARF16D*, *MedARF16E*, *MedARF10A*, and *MedARF10B* genes (Figure 4C). These genes have all undergone duplication events, particularly tandem duplication, during the course of evolution. Overall, our study highlights the presence of cis-regulatory elements associated with the AP2/ERF, BHLH, bZIP, and GATA transcription factor families in the upstream regions of the ARF gene family and various miRNAs in *M. dodecandrum*.

### 2.6. Analysis and Validation of Transcriptome Expression Profiles in Different Organs

In the early stages of our research project, we obtained transcriptome data from various organs of *M. dodecandrum*, including different nutritional organs (leaves, roots, stems), different parts of reproductive organs (flower, stamen, pistil, sepal, petal), and different stages of fruit development (little fruit, medium fruit, large fruit). We conducted an analysis of the expression levels of *MedARFs* in the transcriptome. Based on the transcriptome data, the expression patterns of the 27 ARF genes in *M. dodecandrum* were analyzed (Figure 5A). Among all the 27 *MedARF* genes, the expression levels of the *MedARF2*, *MedARF5B*, *MedARF16C*, and *MedARF16E* genes were very low in all the samples, and in some cases, they were not expressed at all. *MedARF16C* showed significantly higher expression levels compared to the other genes across all the samples. The expression of the *MedARFs* showed distinct tissue specificity, with 3, 11, 7, and 11 genes exhibiting notable expression in leaves, stems, roots, and flowers, respectively. However, certain genes exhibited significant expression levels in specific tissues. For instance, *MedARF17A* and *MedARF17B* exhibited high expression exclusively in flowers; *MedARF16H* was prominently expressed solely in sepals; and *MedARF16D* showed relatively conspicuous expression in petals. Furthermore, the expression pattern of the ARF gene family in *M. dodecandrum* fruits exhibited a specific trend. *MedARF3*, *MedARF7A*, *MedARF9A*, *MedARF9B*, *MedARF9C*, and *MedARF19* showed a gradual decrease in expression levels as the fruits matured, while *MedARF6A*, *MedARF6B*, *MedARF6C*, *MedARF10A*, *MedARF16A*, and *MedARF16F* exhibited an initial increase followed by a precipitous drop.

To ensure the reliability of the transcriptomic data, we selected several *M. dodecandrum* ARF genes (*MedARF3*, *MedARF4*, *MedARF5A*, *MedARF6C*, *MedARF7A*, *MedARF9C*, *MedARF16H*) from different branches of the evolutionary tree based on gene expression profiles. A real-time quantitative polymerase chain reaction (RT-qPCR) was performed to confirm their expressions. These genes showed differential expression levels in various nutritional organs, such as leaves, stems, and roots (Figure 5B). The expression patterns of *MedARF3*, *MedARF4*, *MedARF5A*, *MedARF6C*, *MedARF7A*, and *MedARF16H* were consistent with the transcriptomic data, validating the accuracy and reliability of the transcriptomic data.

### 2.7. Analysis of Gene Expression Profiles in Hormone Treatment

Six ARF genes (*MedARF3*, *MedARF4*, *MedARF5A*, *MedARF6C*, *MedARF7A*, *MedARF16H*) were chosen based on their phylogenetic tree placement and relatively high expression levels in leaves. The leaves of *M. dodecandrum* were treated with 100 μmol/L of indole-3-acetic acid (IAA), a growth-promoting hormone, and 100 μmol/L of naphthalene-1-acetic acid (NPA), a growth inhibitor. RT-qPCR was employed to analyze the relative expression levels of the six candidate *MedARF* genes before and after the treatments at various time points. The results of the IAA treatment (Figure 6A) showed a continuous increase in the expression level of *MedARF3* after treatment, peaking at 12 h for improved clarity. Subsequently, there was a gradual recovery towards the pre-treatment level after 24 h. In contrast, the expression of *MedARF4* was suppressed within the initial 12 h after auxin treatment and gradually returned to the pre-treatment level after 24 h. The remaining genes showed a rapid increase in expression after the IAA treatment, reaching a peak around 6 h and then declining. However, the expression levels of *MedARF6C* and *MedARF7A* gradually returned to normal levels one day after the treatment, while *MedARF9B* and *MedARF16A* showed lower expression levels at 24 h after the IAA treatment compared to before the treatment. The results of the NPA treatment (Figure 6B) showed that *MedARF7* and *MedARF9B* were suppressed after treatment, whereas *MedARF3*, *MedARF4*, *MedARF6C*, and *MedARF16A* remained upregulated. Additionally, the correlation heatmap of gene expression (Figure 6C) revealed a high-positive correlation between *MedARF16A* and *MedARF3*, as well as *MedARF4*, suggesting the possibility of coordinated expression in the expression of these genes.

## 3. Discussion

*Melastoma dodecandrum* is widely used in South China. The investigation of gene expression regulation during the growth of *M. dodecandrum* could provide a scientific foundation for enhancing its variety and facilitating its application in landscaping. ARF transcription factors play important regulatory roles in plant growth. In our study, we analyzed the complete genome data of *M. dodecandrum* and identified 27 ARF family genes. The number of ARF genes in *M. dodecandrum* is higher than that in *A. thaliana* (23) [30], *Oryza sativa* (24) [15], *Citrus reticulata* (19) [31], *and Citrus reticulata* (17) [32], indicating a larger ARF gene repertoire in *M. dodecandrum*. Furthermore, our analysis of homologous gene pairs suggests that the ARF genes in *M. dodecandrum* may have undergone a large-scale duplication event [33] during the course of evolution. And this result led to a massive expansion of *MedARFs*. The physicochemical analysis revealed that the number of MedARF protein amino acids ranged from 319 to 1148. In comparison to the closely related species *Eucalyptus grandis* (593–1119), *M. dodecandrum* displays a significantly higher degree of variation in protein length, exhibiting an excess of more than 50% ARF genes when compared to its counterpart (17). These compelling findings strongly indicate that a remarkable duplication event of ARF members took place during the evolutionary trajectory of the Medinilla genus [32]. The hydrophobicity of MedARF proteins was consistently below 0, ranging from −0.286 to −0.567, indicating that all MedARF proteins are hydrophilic. The subcellular localization prediction showed that the MedARF5B, MedARF6A, MedARF9A, MedARF16A, MedARF16B, and MedARF16C proteins were simultaneously localized in the nucleus and cytoplasm. Based on the analysis of the conserved domains of ARF, it was found that MedARF5B, MedARF6A, and MedARF9A possessed the PB1 domain. This suggests that the presence of the PB1 domain in these genes may lead to the formation of LLPS (liquid–liquid phase separation) condensates during their expression, thereby influencing the localization of ARF [26].

The phylogenetic tree (Figure 1) revealed that the 27 *M. dodecandrum* ARF genes could be distinctly divided into Class I, Class II, and Class III. Apart from the clustering of *MedARF3*, *MedARF4*, and *MedARF9s* in Class I, the remaining MedARF genes clustered in Class II and Class III, displaying considerable divergence [34]. Based on the tertiary structure predictions, the MedARF proteins within the Class I branch exhibited relatively consistent size and morphology, indicating evolutionary conservation. In contrast, the proteins from Class II and Class III showed distinct morphological differentiation, particularly MedARF5A, which displayed significant differences in structure compared to the other proteins. And in combination with the spatial structure occupancy of MedARF proteins (Appendix A), we conclude that the reason for the huge difference between MedARF5A and other proteins may be because it has the shortest sequence length of all the proteins (319). Based on the analysis of the morphology and abundance of *MedARFs* in different branches, we can conclude that during the evolution of MedARF transcription factors, there was a constrained expansion of members within the first class of transcription factors, while the *MedARFs* in the Class II and Class III branches underwent duplication. The chromosomal analysis and sequence analysis of the *MedARFs* (Figure 2) indicated that the unique structure of MedARF5A resulted from a significant deletion of a domain during its evolution, leading to its differentiation from the rest of the group due to its relatively shorter length. Additionally, certain motifs, such as motif 14 and motif 15, were only present in Class III of the *MedARFs* [35]. The complete PB1 domain was found in all the *MedARFs* clustered in Classes I and II, suggesting that the insertion and loss of these motifs played a crucial role in the evolution of the *MedARFs* [36].

Repeated events often lead to the expansion and functional diversification of gene families [37]. From the analysis of repeated events among the *MedARFs* (Figure 3), we found that, except for a few highly conserved genes, 15 repeat events occurred among the 27 *MedARFs*, involving 22 genes. Among them, *MedARF16s* gave rise to multiple members, with a count of eight, which accounts for approximately one-third of the total ARF members in *M. dodecandrum*. Combining our previous studies on *M. dodecandrum*, it can be inferred that the ARF gene family in *M. dodecandrum* actively participated in WGD (whole-genome duplication events), resulting in extensive functional and membership diversification. This process can facilitate coordinated gene expression and the evolution of novel functions and provides raw material for the evolutionary innovation of *M. dodecandrum* through gene dosage effects and functional redundancy.

The detection of miRNA target genes provided further evidence for the functional similarity among several *MedARF16s*. Highly conserved miRNA splice sites (miR160b) [38] were predicted in the gene clusters of *MedARF16C*, *MedARF16D*, and *MedARF16E*, as well as *MedARF10A* and *MedARF10B*. Through the aforementioned studies, it was discovered that these genes had all undergone repeated events, with particular emphasis on the tandem duplication between *MedARF16C* and *MedARF16D*. It is speculated that in the early stages of evolution, the ancestral *MedARF16s* primarily resided on *M. dodecandrum* chromosome 2 and underwent continuous expansion outward. This finding aligns with the clustering pattern of the Class II branch in the evolutionary tree, suggesting that these genes may possess similar functions and have undergone expansion during the course of evolution. The cis-regulatory elements are correlated with gene regulatory functions [39,40]. In this study, we identified a total of 37 transcription factor binding sites in the 2000 bp upstream region of the *MedARFs* promoters, and their distribution among the 27 ARF genes was found to be uneven (Figure 4). The upstream region of the MedARFs is enriched with cis-regulatory elements associated with the AP2/ERF [41], BHLH [42], Bzip [43], GATA, and WRKY [44] transcription factor families (Figure 4A). The AP2/ERF, BHLH, bZIP, GATA, and WRKY transcription factor families may collectively play crucial roles in regulating plant growth and development, including stem elongation and fruit development. Specifically, we observed an abundance of promoter elements related to the BHLH and bZIP families in multiple members of the *MedARF16s* subgroup, indicating that they might be regulated by downstream genes targeted by the BHLH and bZIP transcription factors [45], thereby exhibiting similar regulatory functions. They may co-regulate complex controlling the transcription of *MedARFs* and thereby influencing gene expression during the growth of *M. dodecandrum* [46].

The expression profiles of *MedARFs* in different organs showed that *MedARFs* exhibited certain tissue specificity in different organs (Figure 5A). They were highly expressed in stems and roots. Their abundance of expression in stems and roots suggests a potential involvement of *MedARFs* in auxin transport [47]. In the reproductive organs, eleven genes were highly expressed in flowers and fruits, and some genes showed significant expression only in specific tissues in flowers. For instance, *MedARF17A* and *MedARF17B* were highly expressed only in flowers; *MedARF16H* exhibited high expression specifically in sepals; and *MedARF16D* showed relative prominence in petals. Moreover, the expression of the ARF gene family in *M. dodecandrum* fruit followed a certain pattern. The expression levels of *MedARF6A*, *MedARF6B*, *MedARF6C*, *MedARF10A*, *MedARF16A*, and *MedARF16F* showed an initial increase followed by a sharp decrease. Previous investigations have demonstrated that the floral organs of the Arabidopsis thaliana mutant variant lacking ARF6 maintain a regular count and spatial arrangement during the period of floral blooming. However, these flowers encounter impediments in the subsequent formation of capsules [48]. This observation suggests that these *MedARFs* may have significant involvement in the regulation of fruit development [49].

Auxin is an essential hormone in plant development, influencing embryonic differentiation, plant growth, and overall plant survival [50]. These biological processes cannot be initiated without auxin induction. Current research indicates that ARF TFs play crucial roles in regulating various developmental processes in plants through hormone regulation and response [4]. In this study, the response patterns of ARF transcription factors to auxin in *M. dodecandrum* were examined and analyzed (Figure 6). The results demonstrated that under IAA treatment, *MedARF3*, *MedARF6C*, *MedARF7A*, *MedARF9B*, *and MedARF16A* exhibited a positive correlation with auxin treatment, and their expressions could respond rapidly to auxin within 6 h. This finding is consistent with studies on ARFs in *S. melongena* [51], suggesting that these five genes actively participate in the regulation of *M. dodecandrum* growth in response to auxin. On the other hand, *MedARF4* showed sustained inhibition after treatment, suggesting some antagonistic interactions with specific transcription factors in its expression process following auxin treatment. Additionally, under the treatment of the auxin transport inhibitor NPA, the expression of *MedARF3*, *MedARF4*, *MedARF6C*, and *MedARF16A* continuously increased, while *MedARF7A* and *MedARF9B* were suppressed. The implications of these findings suggest that these MedARFs may play a crucial role in regulating plant growth and development in *M. dodecandrum* by responding to changes in auxin [52].

## 4. Materials and Methods

### 4.1. The Plant Material and Auxin Treatment of M. dodecandrum

In this study, *M. dodecandrum* plants grown in the forest orchid garden of Fujian Agriculture and Forestry University (coordinates: 119°14′35.88″ E, 26°05′2.48″ N) were selected as the research subjects. The *M. dodecandrum* genome data and floral organ transcriptome data used in this study are derived from previous work by Hao et al. [25]. To investigate the expression patterns of the *MedARF* gene family, we obtained samples from three different organs of mature *M. dodecandrum* plants, namely leaves, stems, and roots. Additionally, hormone treatments were conducted on the naturally grown whole plants according to a method described by Zhou et al. [28]. The treatments in this study included the application of exogenous auxin indole-3-acetic acid (IAA, 100 μM) and a growth hormone transport inhibitor, naphthylphthalamic acid (NPA, 100 μM) solution. The plants were evenly sprayed with a misting device until the surfaces were thoroughly moistened, avoiding droplet condensation. Prior to the treatments, leaf samples were collected at specific time points. For the IAA treatment, leaf samples were collected 6, 12, and 24 h after the treatment. In the case of the NPA treatment, leaf samples were collected 6 and 24 h after the treatment. The collected samples were individually placed into sterile and enzyme-free cryotubes with a volume of 2.5 mL. The mixed samples were then stored for subsequent experiments.

### 4.2. Identification and Phylogenetic Analyses of ARF Genes in M. dodecandrum Genome

The *M. dodecandrum* genome data used in this study follow Hao et al. [25]. To obtain the AtARF protein sequences, we downloaded 23 AtARF sequences from TAIR v10 (http://www.arabidopsis.org/, accessed on 5 October 2023). The obtained sequences were subjected to a BLASTP search, and the results were further analyzed using the TBtools software (v2.010) [53]. To refine the ARF gene sequence matches from the BLASTP search, we employed the Hidden Markov Model (HMM) with B3 (PF02362) and Auxin resp (PF06507) domains through the TBtools software (v2.010). In addition, we used the CD-search function of NCBI to screen the above preliminarily identified *MedARF* genes, and the search of the database was set to CDD v3.20-59693 PSSMS. The maximum number of hits was set to 500, and the rest was set as default. Results: 27 ARF genes of *M. dodecandrum* were screened. These steps were crucial to ensuring the accurate identification of ARF gene sequences. We obtained the ARF protein sequences of *L. chinensis* [16] and *O. sativa* [15] from the literature, constructed a phylogenetic tree by comparing them with the ARF protein sequences of *A. thaliana* and *M. dodecandrum*, and named the identified *MedARFs* based on sequence alignment results.

### 4.3. Tertiary Structure and Physicochemical Properties Analyses of MedARFs

We employed the Expasy website (http://web.expasy.org/compute/, accessed on 17 October 2023) to predict the physicochemical properties of the identified MedARF proteins [54]. Additionally, the WoLF PSORT tool (https://wolfpsort.hgc.jp/, accessed on 17 October 2023) was utilized to forecast the cellular localization of these proteins [55]. Moreover, for the prediction of the tertiary structures of the *MedARFs*, we utilized the Protein Structure Prediction tools (AlphaFold2.3.2) (https://colab.research.google.com/github/deepmind/alphafold/blob/main/notebooks/AlphaFold.ipynb, accessed on 16 November 2023) [56]. In addition, we used SOMPA (https://npsa-pbil.ibcp.fr/cgi-bin/npsa_automat.pl?page=npsa_sopma.html, accessed on 21 December 2023) to analyze the secondary structure of MedARF proteins, with the output width set to 70 and the number of conformational states set to 4 (Helix, Sheet, Turn, Coil).

### 4.4. Analysis of MedARFs’ Conserved Domains, Structures, and Motifs

Using the online website MEME (https://meme-suite.org/meme/doc/meme.html, accessed on 20 November 2023) [57], we analyzed the motifs of *M. dodecandrum*. The number of motifs was set to 15, while the other parameters were maintained at their default values. ESPrip (https://espript.ibcp.fr/ESPript/cgi-bin/ESPript.cgi, accessed on 20 November 2023) was used to show the amino acid arrangements and secondary structures. Furthermore, gene structures and conserved motifs were visualized using the GeneStructureView (Advanced) plugin in the TBtools software (v2.010) [53].

### 4.5. MedARF Gene Duplication Events and Selection Pressure Analysis

Based on the genome annotation, we located the genomic locations of *MedARFs* in the Dioscorea genome. To identify tandem and segmental duplications of MedARF in the *M. dodecandrum* genome, we employed MCscanX with a significance threshold of e-value ≤ 1 × 10^−10^ for the matches [58]. For the analysis and visualization of the distribution of *MedARFs* on chromosomes, gene density, and their homology relationships, we utilized the Advanced Circos function in TBtools software (v2.010).

### 4.6. Predict of MedARFs’ Micro-RNA Editing and cis-Acting Elements

To explore the interaction between miRNAs and their target *MedARFs*, we conducted bioinformatics and prediction analyses using the web-based psRNA Target Server (https://www.zhaolab.org/psRNATarget/analysis, accessed on 19 November 2023) [59]. For the analysis, we set the expected value to 4 while keeping the remaining parameters at their default values. Subsequently, we conducted an alignment between the identified genes and the miRNAs of *A. thaliana*, and the results were appropriately labeled. For the extraction of the upstream 2000 bp sequences of the identified 27 *MedARFs*, we employed TBtools software (v2.010). These sequences were then subjected to cis-regulatory element prediction through PlantPAN 4.0 (http://plantpan.itps.ncku.edu.tw/plantpan4/promoter_multiple.php, accessed on 22 November 2023) [60]. The predicted results were further analyzed and visualized using Excel (2021).

### 4.7. Expression Pattern Analysis of MedARFs in Different Tissues and Hormone Treatments

To investigate the potential involvement of ARF genes in different organs of *M. dodecandrum*, we analyzed the expression patterns of 27 ARF genes in various organs. The transcriptome data were obtained from the Genome Sequence Archive (https://ngdc.cncb.ac.cn/gsa/, accessed on 15 November 2023) number: CRA004347 [25]. The leaves, stems, roots, flowers (including stamen, pistil, sepal, petal), and fruits (encompassing expansion period, coloring period, and full-ripening period) were collected from *M. dodecandrum*. The RNAprep Pure Plant Plus Kit from TIANGEN (https://www.tiangen.com, accessed on 18 November 2020) was used to extract total RNA. Each sample was analyzed with three biological replicates. Refer to Hao et al. [25] for methods for the alignment and assembly of *M. dodecandrum’s* transcriptome data. Illumina RNA-Seq libraries were constructed and sequenced on the HiSeq 2500 system. After sequencing, the clean reads from each sample were mapped to the assembled genome, and HTseq-count was used to obtain read counts for each gene. FPKM counts were normalized to estimate gene expression levels. Data visualization was performed using TBtools-ll software (v2.010).

The relative gene expression levels were calculated using the 2^−ΔΔCt^ method based on the RT-qPCR data. The isolation of total RNA, reverse transcription, and RT-qPCR was performed according to a method described by Zhou et al. [28]. The primer sequences for the target genes and the internal references can be found in the Appendix A. To ensure the reliability of the transcriptome data, we combined the RT-qPCR data of *MedARFs* in different nutritional organs with the transcriptome data for a joint analysis using GraphPad Prism 9. Furthermore, the RT-qPCR data obtained from *M. dodecandrum* treated with auxin were subjected to a one-way analysis of variance (ANOVA) for a statistical analysis. The significance levels were indicated as * for *p* < 0.05 and ** for *p* < 0.01. Origin (v2021) software was used for data visualization and analysis. Furthermore, for the RT-qPCR data obtained after the IAA and NPA treatments, we conducted a gene expression correlation analysis of the *MedARFs* using the correlation heatmap function of chiplot (https://www.chiplot.online/, accessed on 25 November 2023).

## 5. Conclusions

In this study, we identified 27 members of the ARF genes in *M. dodecandrum*. According to the results of the phylogenetic analysis, the *MedARFs* were classified into Classes I-III. The *MedARFs* in the Class I branch were relatively conserved, while the members of the Class II and Class III branches had undergone a lot of duplication events. The protein structure analysis, sequence alignment, investigation of structural features, prediction of miRNA binding sites, and analysis of repeat events further confirm this result. And we found that the occurrence of a substantial fragment deletion in *MedARF5A* within Class I leads to its relatively autonomous status, whereas the repeated duplications in *MedARF16* of Class III establish it as the biggest bunch. The results of the expression profile revealed that the expression levels of *MedARF6s* and *MedARF16s* showed an initial increase followed by a sharp decrease at different stages of fruit development. And the expression of these *MedARFs* positively responds to changes in exogenous auxin. The above results illustrate that *MedARFs* may play important roles in regulating fruit development and auxin response in *M. dodecandrum*. These findings provide insights into the evolution of ARF transcription factors and their regulation of auxin responses in *M. dodecandrum*, uncovering potential research directions.

## Figures and Tables

**Figure 1 ijms-25-00806-f001:**
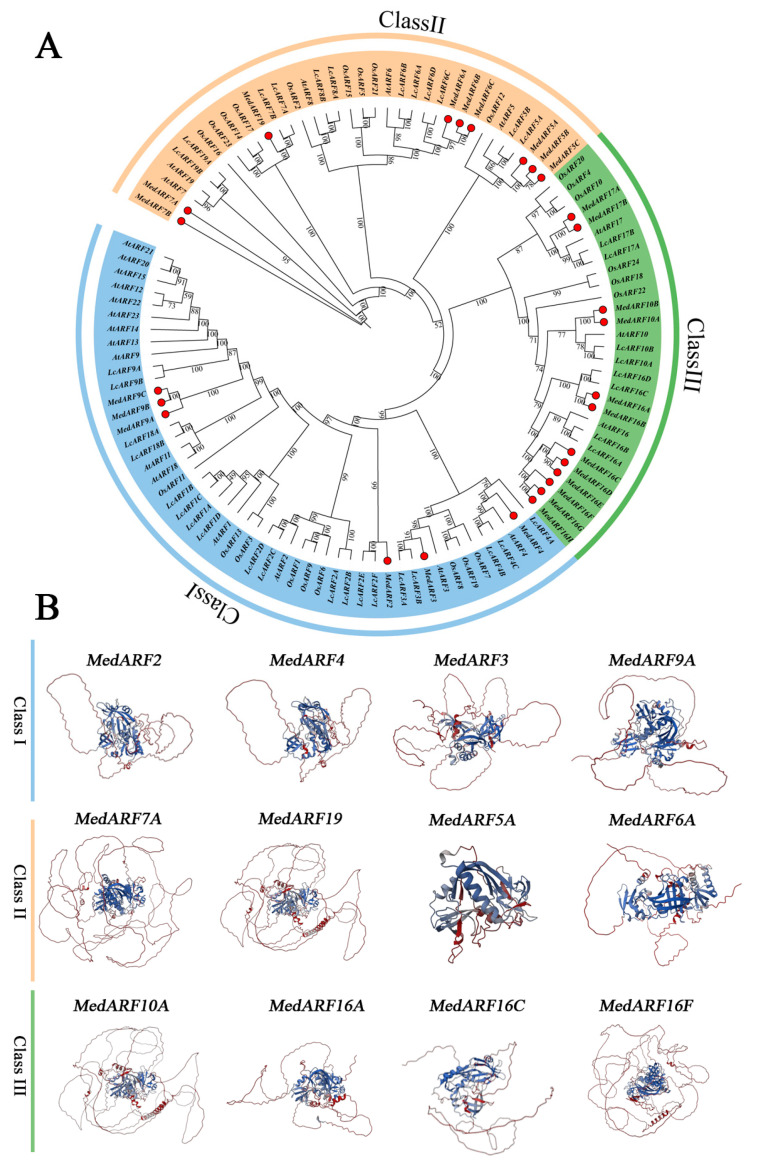
Phylogenetic analysis and tertiary structure of MedARF proteins. (**A**) Phylogenetic analysis of ARFs from *M. dodecandrum*, *A. thaliana*, *L. chinensis*, and *O. sativa*. Numbers on the nodes indicate the credibility values of each clay. The red dots indicate MedARF proteins of *M. dodecandrum*. (**B**) Prediction analysis of tertiary structure of MedARF proteins. Three subgroups are shown as Class I, II, and III. Different colored lines and red circles indicate the confidence of the protein structure, increasing from red to blue.

**Figure 2 ijms-25-00806-f002:**
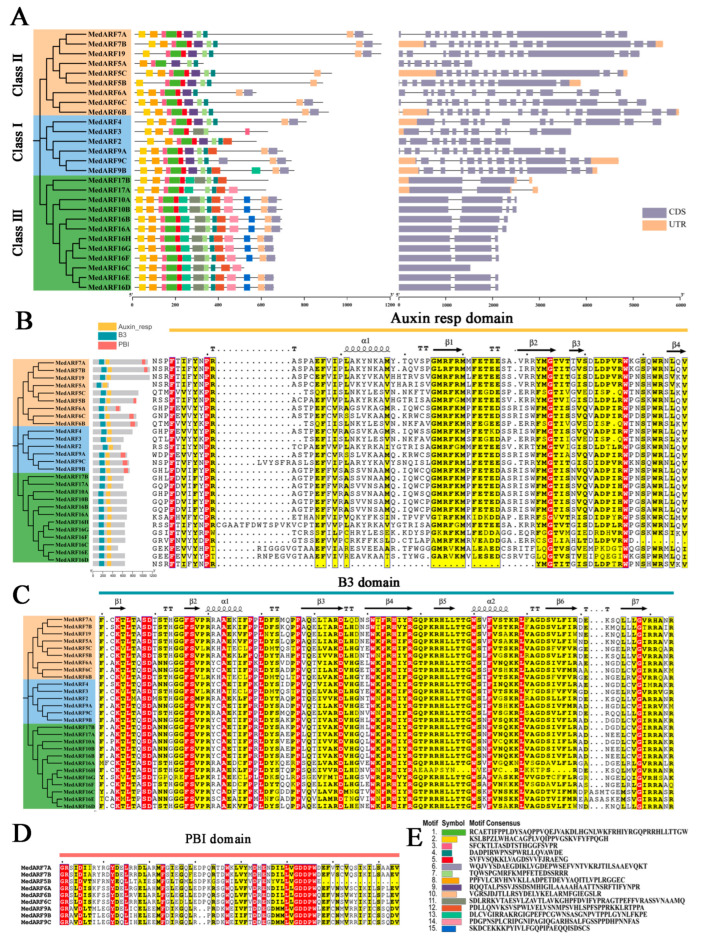
Classification and sequence analysis of *MedARFs*. (**A**) Genetic structure and motif distributions of *MedARFs*. (**B**) The division of the domain and multiple sequence alignment of ARF domains in MedARF proteins. (**C**) Multiple sequence alignment of B3 domains in MedARF proteins. (**D**) *MedARF* genes with PB1 motif. (**E**) Amino acid sequence of each motif. Three subgroups are shown as Class I, II, and III.

**Figure 3 ijms-25-00806-f003:**
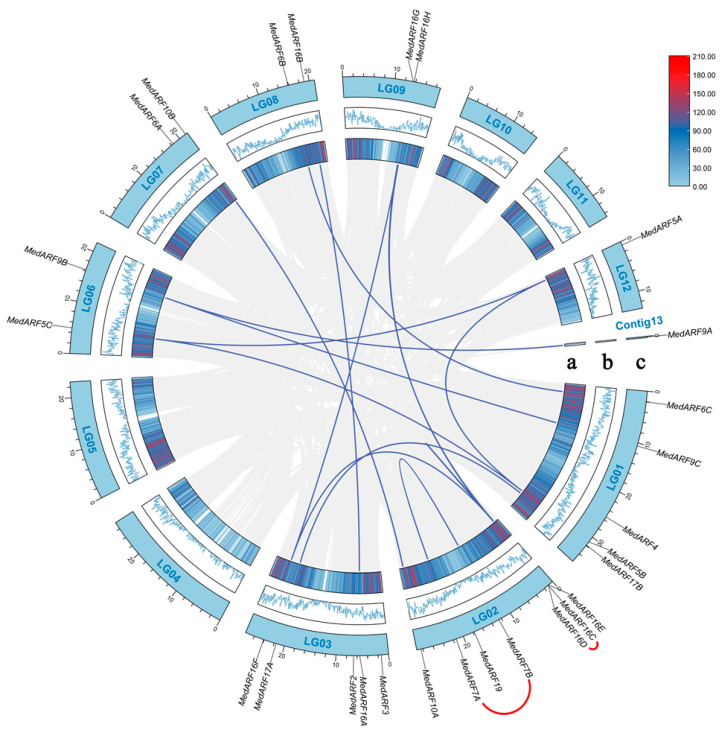
Chromosomal locations of *MedARFs* and the occurrence of repeated events between them. a: line colors indicate gene density. Deep red indicates a higher gene density, while light blue indicates a lower gene density. b: the blue line represents gene density. The higher the height of the line segment, the higher the gene density within a given unit distance. c: LG01–LG12 represents *M. dodecandrum*’s 12 chromosomes and contig13. Homologous gene pairs are connected by blue line segments. The red lines outside the circular diagram specifically highlight the genes that undergo tandem gene duplications. The gray lines indicate all the duplication events that have occurred within the genome of *M. dodecandrum*.

**Figure 4 ijms-25-00806-f004:**
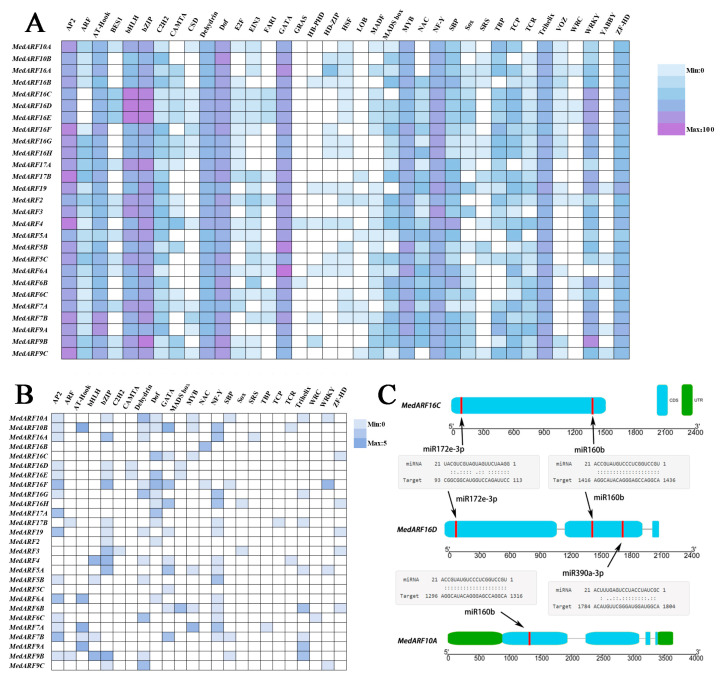
Analysis of the cis-regulatory elements and targets of specific miRNAs for *MedARF* genes. (**A**) Heatmap analysis of cis-acting elements of 2 kb promoter regions of *MedARF* genes. (**B**) Heatmap analysis of cis-acting elements of 35 bp promoter regions of *MedARF* genes. (**C**) Prediction of targets for some miRNAs. This view displays partial regulatory roles of miRNAs and *MedARF* TFs. Blue parts show the coding region of *MedARFs*. Green parts show the untranslated Region. Red lines show the splicing sites.

**Figure 5 ijms-25-00806-f005:**
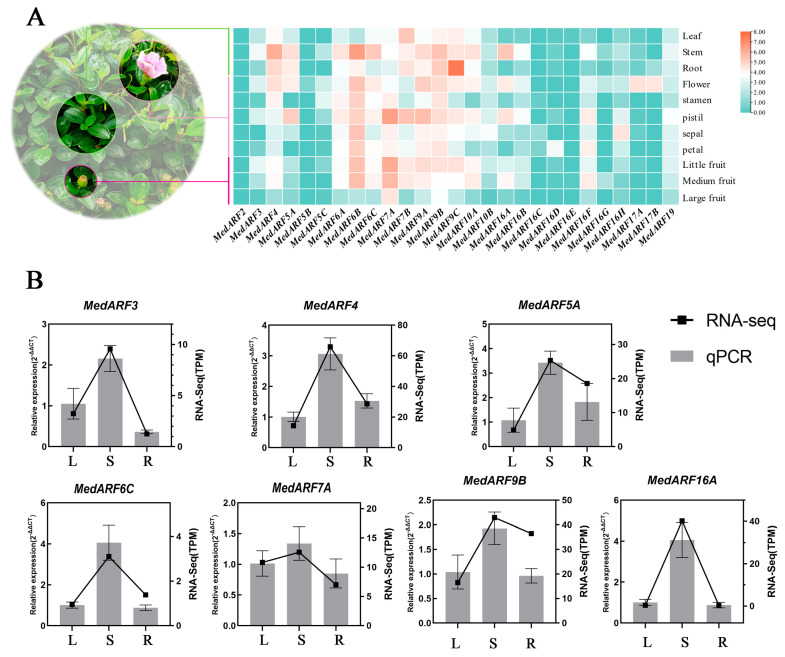
Expression profile analysis of different *MedARFs* in *M. dodecandrum.* (**A**) Heatmap of the expression patterns of the ARF gene family in *M. dodecandrum*. Little fruit: encompassing expansion period; Medium fruit: coloring period; Large fruit: full-ripening period. The color scale represents the normalized log^2^−transformed counts per million kilobases of reading. (**B**) The relative expression of *MedARFs* in leaf, stem, and root vegetative organs and transcriptome correlation analysis. L: leaf. S: stem. R: root. RT-qPCR results correspond to the left ordinate. RNA-seq results correspond to the right ordinate.

**Figure 6 ijms-25-00806-f006:**
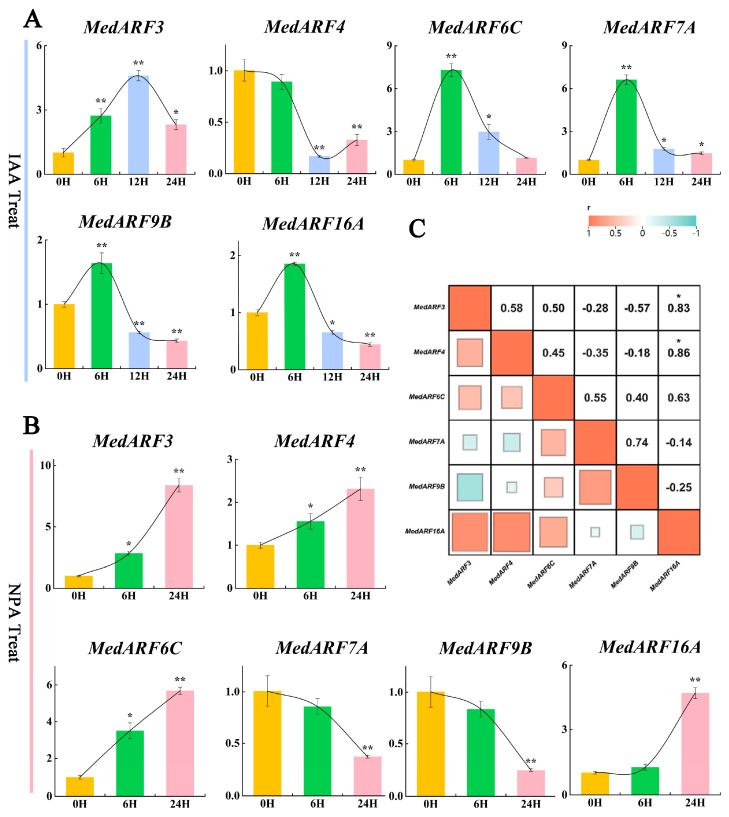
*MedARF* expression 0 h, 6 h, 12 h, and 24 h after 100 μmol/L of IAA treatment and 0 h, 6 h, and 24 h after 100 μmol/L of NPA treatment. (**A**) The expression of *MedARFs* with IAA treatment; (**B**) The expression of *MedARFs* with NPA treatment. Relative transcript levels are calculated by RT-qPCR with MedActin1 as a standard. Data are means ± SE of three separate measurements based on *t*-test, taking as *p* < 0.05 * and *p* < 0.01 as **. (**C**) Heatmap shows the correlation of gene expressions between the two treatments. The more squares there are, the stronger the correlation, with red representing a positive correlation and blue representing a negative correlation.

## Data Availability

The original genome sequences described in this article have been submitted to the National Genomics Data Center (NGDC, https://ngdc.cncb.ac.cn, accessed on 16 August 2023) under accession number PRJCA005299; raw transcriptome data were stored at GSA, portion accession number: CRA004347. All data generated or analyzed during this study are included in this published article or Appendix A and are also available from the corresponding author on reasonable request. Comparative data were obtained from the Plant Transcription Factor Database 5.0 website (http://planttfdb.gao-lab.org/, accessed on 16 August 2023).

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
