# Peer review of "A Comprehensive Analysis of Auxin Response Factor Gene Family in Melastoma dodecandrum Genome"

_ijms, 2024, doi:10.3390/ijms25020806_

Round 1
Reviewer 1 Report
Comments and Suggestions for Authors
Overall, this manuscript presents a well-conceived study integrating bioinformatic analysis with wet-lab validation to investigate the functions of MedARF genes. Despite its merits, the manuscript requires substantial enhancements in the results description and methods section to enhance clarity and coherence. Below, I've outlined the primary concerns:
1. Figure 1A: The criteria for grouping in the phylogenetic tree are unclear. And, the red dots indicated in the tree lack description in the figure legend.
2. Figure 1B: The orientation of the AlphaGo predicted protein structures should align with the conserved domain (e.g., ARF domain) to be meaningful. The absence of further wet-lab analysis based on the prediction results calls for a more detailed description and appropriate data presentation. Additionally, the accuracy of the prediction should be clearly presented.
3. Result Section 2.1: The manuscript does not specify the criteria of similarity and probability, nor the parameters used in running HMMER or CD-search.
4. Figure 2A: The ARF, PB1, and B3 domains should be labeled on the protein sequence map.
5. Figure 2B: The implications of these results are not clear. A detailed description should be included in the results or discussion section.
6. Result Section 2.4: The terms “13 segments of duplications” and “15 pairs of duplicated genes” are ambiguous. Clarification in the context of the results, along with appropriate markings in the figures, is necessary. Moreover, in line 178, there is an incorrect citation (Table S2 instead of Table S1).
7. Figure 3: The significance of the gray color lines should be explained in the figure legend. The unit for gene density presented in the sidebar needs clarification.
8. Figure 4: The rationale behind analyzing the cis-regulatory segment for MedARF genes is inadequately explained, resulting in a weak logical connection with the preceding sections.
9. Figure 5: The manuscript lacks a description of the transcriptome analysis in the methods section, specifically concerning sequencing output and procedures, including library preparation, sequencing, and data analysis. This information should be comprehensively added to the methods section.
Comments on the Quality of English LanguageModerate editing of English language required
Author Response
On behalf of my co-authors, we sincerely appreciate the opportunity you have provided us to revise and enhance the quality of our article. We have thoroughly considered your valuable suggestions and have made several modifications accordingly. We have exerted our utmost effort to improve the manuscript, and we would like to take this opportunity to briefly explain the changes we have implemented as follows:
1. Figure 1A: The criteria for grouping in the phylogenetic tree are unclear. And, the red dots indicated in the tree lack description in the figure legend.
R: Thank you for your valuable feedback on our manuscript. I have already added a description of the meaning of the red dots in the caption of Figure 1, and I have also continued to describe the clustering results and reference classification in the Results section. Thank you for your constructive criticism.
2.Figure 1B: The orientation of the AlphaGo predicted protein structures should align with the conserved domain (e.g., ARF domain) to be meaningful. The absence of further wet-lab analysis based on the prediction results calls for a more detailed description and appropriate data presentation. Additionally, the accuracy of the prediction should be clearly presented.
R: Thank you for your valuable feedback on our manuscript.The orientation of the protein structures conserved domains I have placed them in (Figure 2 CDE). Additionally, based on your suggestion to address the limitations of the information presented in (Figure 1B), I have added a new table(Tablet S2) as an attachment specifically showcasing the predicted results of the MedARF protein structures to accurately and clearly present the information.Thank you for pointing out the shortcomings of this manuscript.
3.Result Section 2.1: The manuscript does not specify the criteria of similarity and probability, nor the parameters used in running HMMER or CD-search.
R: Thank you for your valuable feedback. We appreciate your opinion and apologize for not specifying the criteria for determining the similarity of MedARFs in the manuscript. In response to your concerns, we will revise the original manuscript to provide additional explanation on the basis of naming MedARFs in section 2.1. Furthermore, specific parameters used when running HMMER or CD-search have not been disclosed for the following reasons: The HMMER search used in this study is a feature of TBtools, and this software feature does not have parameter settings options, hence it was not included in the manuscript. For CD-search, default options were used in this study but were not mentioned in the text, which may cause confusion for readers, and we sincerely apologize for this oversight. Therefore, I will supplement the section 4-2 Materials and Methods with the selection of CD-search parameters to ensure the comprehensibility of the manuscript.
4.Figure 2A: The ARF, PB1, and B3 domains should be labeled on the protein sequence map.
R: Thank you for bringing this to my attention. I appreciate your feedback. I have made the necessary revisions to (Figure 2B) and have now included clear labels for the ARF, PB1, and B3 domains on the protein sequence map.
5.Figure 2B: The implications of these results are not clear. A detailed description should be included in the results or discussion section.
R:Thanks for your reminding, I updated Figure 2B and made a new Figure(Figure S1) of the specific arrangement structure of Amino acid sequence of each motif to give a more detailed description, and supplemented it in the result or discussion section.
6.Result Section 2.4: The terms “13 segments of duplications” and “15 pairs of duplicated genes” are ambiguous. Clarification in the context of the results, along with appropriate markings in the figures, is necessary. Moreover, in line 178, there is an incorrect citation (Table S2 instead of Table S1).
R:Thank you very much for your valuable advice. For the ambiguity of the terms "segment repeats" and "pairs of duplicated genes",we will provide explanations in the Results section and include appropriate markings in the figures. Additionally, we will provide detailed elaboration in the Discussion section to ensure that readers can understand these events clearly. And we have corrected the misreference (Table S2). We will revise the article as soon as possible to ensure the accuracy and clarity of the article.Thanks again for your guidance!
7.Figure 3: The significance of the gray color lines should be explained in the figure legend. The unit for gene density presented in the sidebar needs clarification.
R:Thank you for your feedback. We apologize for the oversight in not explaining the significance of the gray color lines in Figure 3 and not clarifying the unit for gene density in the sidebar. We will revise the figure legend to include an explanation of the gray color lines, highlighting their representation of segmental duplications and tandem gene duplications. Additionally, we will provide a clear clarification of the unit for gene density in the sidebar. Thank you for bringing these points to our attention, and we appreciate your valuable input.
8.Figure 4: The rationale behind analyzing the cis-regulatory segment for MedARF genes is inadequately explained, resulting in a weak logical connection with the preceding sections.
R:Thank you for your valuable feedback. We have revised 2-5 to provide a more comprehensive explanation of the rationale behind analyzing the cis-regulatory segments for MedARF genes. Specifically, we have strengthened the logical connection with the preceding sections by elucidating how the expression regulation of the ARF gene family is interconnected with different transcription factor families. We believe that these enhancements have improved the clarity and coherence of the manuscripts.We appreciate your insights and are confident that the revisions have addressed the concerns raised. Thank you for helping us improve the quality of our manuscripts.
9.Figure 5: The manuscript lacks a description of the transcriptome analysis in the methods section, specifically concerning sequencing output and procedures, including library preparation, sequencing, and data analysis. This information should be comprehensively added to the methods section.
R:Thank you for your feedback. The transcriptome data used in this study is derived from datasets generated by our research group. However, we did not provide a sufficient description of the transcriptome analysis in the manuscript, which may have caused confusion for readers. To address this shortcoming, we plan to briefly outline the referenced transcriptome data and its characteristics to provide readers with a preliminary understanding of the dataset. Additionally, we will include detailed information regarding the transcriptome data analysis in the Materials and Methods section. We apologize for any inconvenience caused and are committed to improving the manuscript. Thank you for your valuable input.
Reviewer 2 Report
Comments and Suggestions for Authors
The work analyze the ARF gene family in Melastoma dodecandrum to identify crucial proteins involved in regulating the response and inhibition of indole-3-acetic acid 22 (IAA). Melastoma dodecandrum is unknown taxa. Why Authors have been chosen this taxa? More about it.
Lines 30-33: unclear in relation to the aims of work.
Line 57: unclear
The Figures 2 and 5 are unreadable. I suggest it should be divided.
Lines 440-444: structure.
Materials and Methods: lack of the preparing of plant material to the analyses and the source of plant material.
Author Response
We would like to extend our sincere appreciation to you on behalf of all the co-authors for granting us the chance to revise and enhance the quality of our article. Your valuable suggestions have been carefully considered, and we have made several changes accordingly. With utmost dedication, we have strived to improve the manuscript by implementing various modifications. In the following section, we will provide a brief explanation of the changes that have been made:
1.The work analyze the ARF gene family in Melastoma dodecandrum to identify crucial proteins involved in regulating the response and inhibition of indole-3-acetic acid 22 (IAA). Melastoma dodecandrum is unknown taxa. Why Authors have been chosen this taxa? More about it.
R: Thank you for your valuable feedback on our manuscript. We chose to study Melastoma dodecandrum as our research subject because it is the only plant within the family Melastomataceae in the order Myrtales that exhibits a creeping growth habit. Therefore, the application of M.dodecandrum, especially in Fujian province, China, is extensive, and research on its growth and development holds significant importance.We hypothesized that auxin response factors (ARFs) may play a unique regulatory role in its growth. By analyzing the ARF gene family in this plant, we aim to uncover the molecular mechanisms underlying auxin response and inhibition, providing new insights into similar processes in other plant species. I will emphasize the rationale behind our choice in the revised manuscript.Thank you for your reminder.
2.Lines 30-33: unclear in relation to the aims of work.
R: Thank you for your review and valuable suggestions.You are right, the description of the work results in the introduction section of the paper is not clear enough(Lines 30-33). Thank you for your feedback.I have made extensive revisions to the introduction section to ensure that readers can better comprehend the purpose of this study. Thank you very much for your input.
3.Line 57: unclear
R: Thank you for your review and valuable suggestions. It is true that this paper did not provide a detailed analysis of the amino acid preferences in the DBD domain for the transcriptional activation and inhibition of ARF transcription factors, which has caused a lack of clarity in the introduction section. After careful consideration, I have decided to remove this sentence to ensure the clarity of the logical flow. Thank you very much for bringing this to my attention.
4.The Figures 2 and 5 are unreadable. I suggest it should be divided.
R:Thank you for your review and valuable suggestions. You are correct that Figures 2 and 5 have issues with multiple images being stitched together, resulting in unreadable content, especially for the motif of ARF genes (Figure 2.B). Based on this, I plan to upload a new attachment specifically for the Logo of conserved motif of MedARFs in M. dodecandrum(Figure S1), making this section clearer and more readable. I will also provide additional explanations in the text to supplement the description.
5.Lines 440-444: structure.
R: Thank you for your review and valuable suggestions. With your reminder, I carefully reviewed the structural issues in Lines 440-444. I have standardized the structure format for website citations in this paragraph and other sections of the Materials and Methods. Additionally, I have also rechecked and standardized the spacing between each paragraph. Thank you for your suggested modifications.
6.Materials and Methods: lack of the preparing of plant material to the analyses and the source of plant material.
R:Thank you for your review and valuable suggestions. You are right, the description of the impact of plant material preparation on analysis and plant material sources in this article is lacking. To address this issue, I have reorganized the relevant content from the Materials and Methods section and included it in section 4-1, along with additional information on the materials used for subsequent data analysis. Thank you very much for pointing this out.
Round 2
Reviewer 1 Report
Comments and Suggestions for Authors
While the authors have addressed some issues with the manuscript, two major concerns remain unresolved, which is necessary to fulfill the data quality standards of IJMS:
-
1. The transcriptome data must be deposited in a public domain (e.g., NCBI SRA), and the analysis strategy needs to be clearly articulated. This critical issue has not been improved in the revised version.
- 2. As addressed in the previous review comment: 'Figure 1B: The orientation of the AlphaGo predicted protein structures should align with the conserved domain (e.g., ARF domain) to be meaningful.' This task is not difficult to accomplish and must be completed to provide evidence that the prediction is reliable. However, this has not been improved in the revised manuscript, raising doubts about the reliability of the structure prediction results.
Before the acceptance of the manuscript, these two issues must be clearly addressed.
Comments on the Quality of English LanguageMinor editing of English language required
Author Response
1.The transcriptome data must be deposited in a public domain (e.g., NCBI SRA), and the analysis strategy needs to be clearly articulated. This critical issue has not been improved in the revised version.
R: Thank you for your valuable feedback on our manuscript. We appreciate your suggestion and have made the necessary revisions to address it. Specifically, we have further supplemented the materials and the analysis strategy in the transcriptome part in lines 537-544. Additionally, we have included the sources of transcriptome data in the Materials and Methods section in lines 534-535 to avoid any potential misunderstanding by the readers. Thank you once again for bringing this to our attention and helping us improve the clarity of our manuscript.
2.As addressed in the previous review comment: 'Figure 1B: The orientation of the AlphaGo predicted protein structures should align with the conserved domain (e.g., ARF domain) to be meaningful.' This task is not difficult to accomplish and must be completed to provide evidence that the prediction is reliable. However, this has not been improved in the revised manuscript, raising doubts about the reliability of the structure prediction results.
R: Thank you for your valuable feedback on our articles. Your comments are of great significance to the protein structure part of my manuscript. I extracted the B3 and Auxin resp domains of MedARF7A, MedARF4 and MedARF17A at the base of the three branches respectively. We use AlphaGo to make predictions about its tertiary structure and provide detailed supplementary information in lines 194-156. We aligned the predicted protein structure direction with the conserved domain in lines 171-174 and (Figure 2B-C) to further enhance the reliability of the structural prediction results. Thank you for your advice.
Round 3
Reviewer 1 Report
Comments and Suggestions for Authors
The report, while not fully meeting my expectations, is sufficient for the current needs. I have reservations about certain aspects, but it will be accepted as is. Further improvement would be beneficial, but I am not insisting on revisions at this point.